# Microbiological, Clinical, and PK/PD Features of the New Anti-Gram-Negative Antibiotics: β-Lactam/β-Lactamase Inhibitors in Combination and Cefiderocol—An All-Inclusive Guide for Clinicians

**DOI:** 10.3390/ph15040463

**Published:** 2022-04-12

**Authors:** Luigi Principe, Tommaso Lupia, Lilia Andriani, Floriana Campanile, Davide Carcione, Silvia Corcione, Francesco Giuseppe De Rosa, Roberto Luzzati, Giacomo Stroffolini, Marina Steyde, Giuliana Decorti, Stefano Di Bella

**Affiliations:** 1Clinical Pathology and Microbiology Unit, “San Giovanni di Dio” Hospital, I-88900 Crotone, Italy; luigi.principe@gmail.com; 2Unit of Infectious Diseases, Cardinal Massaia Hospital, I-14100 Asti, Italy; tommaso.lupia89@gmail.com (T.L.); francescogiuseppe.derosa@unito.it (F.G.D.R.); 3Clinical Pathology and Microbiology Unit, Hospital of Sondrio, I-23100 Sondrio, Italy; lilia.andriani@asst-val.it; 4Department of Biomedical and Biotechnological Sciences, Section of Microbiology, University of Catania, I-95123 Catania, Italy; f.campanile@unict.it; 5Laboratory of Microbiology and Virology, IRCCS San Raffaele Scientific Institute, I-20132 Milan, Italy; carcione.davide@hsr.it; 6Infectious diseases Unit, Department of Medical Sciences, University of Torino, I-10124 Torino, Italy; silvia.corcione@unito.it (S.C.); giacomo.stroffolini@gmail.com (G.S.); 7Clinical Department of Medical, Surgical and Health Sciences, University of Trieste, I-34149 Trieste, Italy; roberto.luzzati@asugi.sanita.fvg.it (R.L.); m.steyde@gmail.com (M.S.); stefano932@gmail.com (S.D.B.); 8Institute for Maternal and Child Health–IRCCS Burlo Garofolo, I-34137 Trieste, Italy

**Keywords:** β-lactams, β-lactamase inhibitors, cefiderocol, pharmacokinetics, pharmacodynamics

## Abstract

Bacterial resistance mechanisms are continuously and rapidly evolving. This is particularly true for Gram-negative bacteria. Over the last decade, the strategy to develop new β-lactam/β-lactamase inhibitors (BLs/BLIs) combinations has paid off and results from phase 3 and real-world studies are becoming available for several compounds. Cefiderocol warrants a separate discussion for its peculiar mechanism of action. Considering the complexity of summarizing and integrating the emerging literature data of clinical outcomes, microbiological mechanisms, and pharmacokinetic/pharmacodynamic properties of the new BL/BLI and cefiderocol, we aimed to provide an overview of data on the following compounds: aztreonam/avibactam, cefepime/enmetazobactam, cefepime/taniborbactam, cefepime/zidebactam, cefiderocol, ceftaroline/avibactam, ceftolozane/tazobactam, ceftazidime/avibactam, imipenem/relebactam, meropenem/nacubactam and meropenem/vaborbactam. Each compound is described in a dedicated section by experts in infectious diseases, microbiology, and pharmacology, with tables providing at-a-glance information.

## 1. Introduction

The epidemiology of infections sustained by multidrug-resistant Gram-negative bacteria is rapidly evolving. New drugs are available or are on the horizon. Most are combinations of a β-lactam and a β-lactamase inhibitor. One part is the antibiotic cefiderocol that has a peculiar antibacterial mechanism of action. Dispensing of such an armamentarium requires in-depth knowledge of their microbiological spectrum of activity, pharmacokinetic/pharmacodynamic (PK/PD) properties, and clinical study results. Herein, we aimed to summarize the new antibacterial molecules in order to help clinicians in choosing the most appropriate drug according to the type of patient (e.g., obese, critically ill, nephropathic), the type of bacterium (e.g., non-fermenting Gram-negative), and the site of infection (e.g., pneumonia, skin and soft tissue, bloodstream infections). The following molecules are described: aztreonam/avibactam, cefepime/enmetazobactam, cefepime/taniborbactam, cefepime/zidebactam, cefiderocol, ceftaroline-fosamil/avibactam, ceftolozane/tazobactam, ceftazidime/avibactam, imipenem/relebactam, meropenem/nacubactam, and meropenem/vaborbactam (Figure 1).

## 2. Aztreonam/Avibactam

Aztreonam is an old antibiotic approved by the Food and Drug Administration (FDA) and the European regulatory authorities in 1986. Its clinical use was strongly limited by the spread of extended-spectrum β-lactamase (ESBL) and AmpC-type determinants. Of note, metallo-β-lactamases (MBLs) are able to hydrolyze all β-lactams except for the monobactam aztreonam. However, due to the frequent co-production of class A β-lactamases or AmpC-type determinants within MBL-producing Gram-negatives, aztreonam remains active only in one-third of cases [1]. For this reason, combining aztreonam with avibactam could represent a good antimicrobial strategy. A single product formulation of aztreonam/avibactam is currently under development in phase 3 studies for the treatment of MBL-sustained infections. Aztreonam/avibactam has antimicrobial activity against carbapenemase-producing *Enterobacterales*, *P. aeruginosa* (including isolates producing *Klebsiella pneumoniae* carbapenemase, KPC; Verona integron-encoded metallo-β-lactamase, VIM; imipenemase, IMP; New Delhi metallo-β-lactamase, NDM; and oxacillinase, OXA-48), and *Stenotrophomonas maltophilia* [2,3]. No antimicrobial activity has been reported against *A. baumannii* (no inhibition of *A. baumannii* OXA-type enzymes). Resistance in *P. aeruginosa* has been associated with impermeability (porin loss), the production of AmpC-type (*Pseudomonas*-derived cephalosporinase 1; PDC) variants, OXA enzymes (other than OXA-48), or hyperexpression of efflux systems, while resistance in *Enterobacterales* could be associated with a specific amino acid insertion (12 bp duplications) in PBP3 determinants causing a reduction in affinity for aztreonam [2] (Table 1). For antimicrobial susceptibility testing purpose, the concentration of avibactam is fixed at 4 mg/L [4]. No clinical breakpoint (CLSI, EUCAST, or FDA) has been approved for this combination. An EUCAST epidemiological cut-off (ECOFF) value has not been assigned.

**Figure 1 pharmaceuticals-15-00463-f001:**
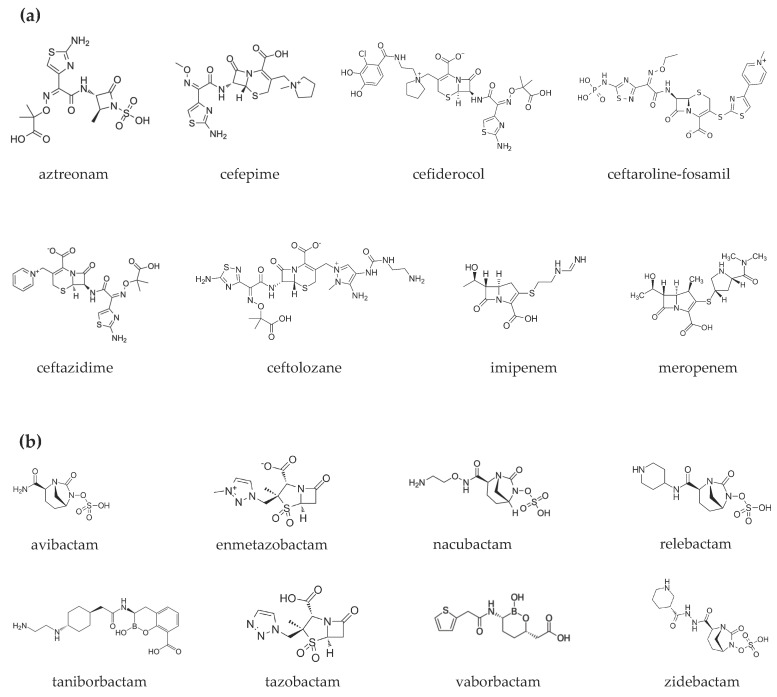
Chemical structures of (**a**) β-lactams and (**b**) β-lactamase inhibitors.

Currently, seven clinical trials on aztreonam/avibactam are registered: five are completed and two are recruiting. The efficacy of the combination is being tested in patients with bloodstream infections (BSIs), complicated intra-abdominal infections (cIAIs), complicated urinary tract infections (cUTIs), hospital-acquired pneumonia (HAP), and ventilator-associated pneumonia (VAP). In contrast, aztreonam/avibactam safety is more generally being evaluated in patients with serious or complicated bacterial infections [5]. Recently, a phase 2 trial was published: 34 patients with cIAIs were treated for 5–14 days with aztreonam/avibactam + metronidazole. No patients had either ESBL or MBL-positive isolates. Patients were divided into three cohorts: (1) 500/137 mg, followed by 1500/410 mg every 6 h; (2) 500/167 mg, followed by 1500/500 mg every 6 h; and (3) extension of exposure at the higher dose regimen. The most common adverse events were hepatic enzyme increases (26%) and diarrhea (15%). Clinical cure rates at the test-of-cure visit were 59% overall [6]. Data from this study supported the regimen selected for the phase 3 trial (500/167 mg, followed by 1500/500 mg every 6 h) (Table 2).

In the recently published phase 2 clinical trial, aztreonam showed, in the first cohort, a geometric mean volume of distribution (Vd) of 20.0 L, 16.9% (geometric coefficient of variance) and a clearance (Cl) of 6.4 L/h (35.4%), while for avibactam, a Vd of 26.0 L (22.0%) and a Cl of 10.1 L/h (42.6%) was described. Similar data were obtained in the other two cohorts [6]. In patients with cIAI, avibactam’s Cl was lower, while avibactam and aztreonam Vd were higher than in healthy volunteers, as expected in critical patients, due to changes in protein levels, extracellular fluids, and blood volume [6], as already described for avibactam Vd in critical patients with comorbidities and burns [7]. Further PK studies on the combination have not yet been conducted, despite clinical experiences in various infections [2]. Studies on avibactam show that the drug diffuses into epithelial lung fluid (ELF) with concentrations around 30% of those in plasma [8,9]. Instead, the blood–brain barrier represents an obstacle to the diffusion [10] (Table 3 and Table 4).

## 3. Cefepime/Enmetazobactam

Enmetazobactam is a new β-lactamase inhibitor similar in structure to tazobactam, with increased bacterial cell penetration and potency. Similar to tazobactam, enmetazobactam inhibits CTX-M, TEM, SHV, and other class A β-lactamases (except for KPC), but does not inhibit class B and D β-lactamases and carbapenemases. Enmetazobactam alone does not exhibit inhibitory activity against Gram-negative bacteria. The cefepime/enmetazobactam combination is active in vitro against ESBL- and AmpC-producing *Enterobacterales* and *P. aeruginosa* [51]. The in vivo efficacy of cefepime/enmetazobactam was demonstrated using a mouse model of septicemia, indicating the ability of enmetazobactam to significantly enhance the therapeutic efficacy of cefepime in vivo. This new combination represents a potential treatment alternative, contributing to “carbapenem sparing” strategies for infections caused by ESBL-producing *Enterobacterales* [52]. No antimicrobial activity was reported against *A. baumannii* and *S. maltophilia* [53] (Table 1). For antimicrobial susceptibility testing purpose, the concentration of enmetazobactam is fixed at 8 mg/L. No clinical breakpoint (CLSI, EUCAST, or FDA) has been approved for this combination. An EUCAST epidemiological cut-off (ECOFF) value has not been assigned [4].

From a clinical point of view, cefepime/enmetazobactam has been evaluated in two phase 1 (NCT03680352 and NCT03685084), one phase 2 (NCT03680612), and one phase 3 trial (NCT03687255) [54,55,56,57]. The recent ALLIUM trial compared cefepime/enmetazobactam (2/0.5 g every 8 h intravenously (i.v.)) to piperacillin-tazobactam (4.5 g every 8 h, i.v.) in patients with cUTIs, including acute pyelonephritis [56]. The baseline participants numbered 1034 (516 in the cefepime/enmetazobactam group and 518 in the piperacillin/tazobactam group). The proportion of patients in the microbiological modified intention-to-treat (m-MITT) population that achieved overall treatment success was 79% (n = 273) and 59% (n = 196) in the cefepime/enmetazobactam and piperacillin-tazobactam groups, respectively [55]. In addition, cefepime/enmetazobactam displayed a higher proportion of patients in the m-MITT population with clinical cure compared with piperacillin/tazobactam (92% vs. 89%), alongside a tolerable safety profile [56]. Of note, a higher rate of *Clostridioides difficile* infections was reported in the cefepime/enmetazobactam group (13% vs. 0%) [56] (Table 1).

The PK profile of enmetazobactam together with cefepime has been evaluated in a mouse septicemia model; in this model, the profile of enmetazobactam mirrors that of cefepime, and an *f*T > MIC of 40–60% for cefepime and the time above a free threshold drug concentration of 20% (*fT* > Ct) for enmetazobactam remain the PK/PD indices predictive of efficacy [14]. A clinical trial on the PK in ELF and tissue penetration has recently been completed. Healthy volunteers were treated with 2 g cefepime/1 g enmetazobactam every 8 h. At steady state, the area under the curve (AUC)_(0–24)_ plasma/AUC_(0–24)_ ELF ratio was 61% (± 29) for cefepime and 53% (±21) for enmetazobactam. The study shows that both drugs diffuse similarly in ELF, providing evidence for the potential role of the association with nosocomial pneumonia [15]. Two other studies on healthy volunteers [55] and on patients with renal insufficiency are ongoing [55,56] (Table 3 and Table 4).

## 4. Cefepime/Taniborbactam

Taniborbactam is a highly potent broad-spectrum boronate derivative β-lactamase inhibitor that acts as an irreversible, covalent inhibitor of serine β-lactamases and as a competitive inhibitor of MBLs. For this reason, taniborbactam presents a broad spectrum of activity, including all four Ambler classes of β-lactamase enzymes, especially the more clinically relevant B1 subclass of MBL (VIM- and NDM-type enzymes). It has excellent penetration of the outer membrane of Gram-negatives [58]. Taniborbactam combined with the cephalosporin cefepime lowered the MIC of cefepime against ESBL-, AmpC-, and carbapenemase-producing isolates. In contrast, isolates of *bla*_NDM-5_-producing *E. coli* were reported, presenting MIC values > 8 mg/L for cefepime/taniborbactam. In addition, penicillin-binding protein (PBP)3 mutations may be the main reason for higher MICs of the combination among NDM-producing *E. coli* [59]. The combination showed antimicrobial activity against *S. maltophilia*, but not against *A. baumannii* [60] (Table 1). For antimicrobial susceptibility testing purposes, the concentration of taniborbactam is fixed at 4 mg/L [4]. No clinical breakpoint (CLSI, EUCAST, or FDA) has been approved for this combination. An EUCAST epidemiological cut-off (ECOFF) value has not been assigned.

Currently no human studies on cefepime/taniborbactam have been published. However, in a recent study, Lasko et al. assessed the efficacy of the combination in a neutropenic murine cUTI model. The authors used dosing regimens resembling human exposure to 2/0.5 g every 8 h. Eighteen cefepime-resistant clinical isolates (ESBL, AmpC, KPC, OXA-48) were tested. Cefepime/taniborbactam exhibited robust killing of kidney bacteria (until MIC of 32 mg/l) [60]. Two trials are being conducted in humans: one is a study assessing safety in healthy subjects and the other is a phase 3, randomized, double-blind noninferiority study (currently recruiting) aimed at evaluating cefepime/taniborbactam vs. meropenem for the treatment of cUTI. The primary outcome is a composite of microbiological eradication and symptomatic clinical success at test-of-cure [61]. Results are not yet available (Table 2).

As mentioned above, preclinical studies show that cefepime and cefepime/taniborbactam concentration–time profiles are comparable in the murine model and in humans. In the neutropenic murine thigh infection model, Abdelraouf et al. showed that the best PK/PD index remains *f*T > MIC of 50% for the cephalosporin and the *f*AUC_24_/MIC for the β-lactamase inhibitor [17]. No human studies on the PKs of the combination cefepime/taniborbactam have been published and the results of the two trials currently ongoing have not been reported yet [61]. Dowell et al. have evaluated the safety and PKs of single and multiple doses of taniborbactam in human volunteers. The study shows that multiple doses (750 mg every 8 h) result in an AUC of 139.5 (±21.6) h * ng/mL, a Vd of 37.4 (±19.9) L, a half-life (t ½) of 4.7 (±15.4) h, and a renal Cl of 5.6 (±2.1) L/h [18]. The mean fraction of the drug excreted unchanged in urine was 92.4% (±10.2). Cefepime is also excreted unchanged in the urine [16], and the combination is being studied in cUTIs (Table 3 and Table 4).

## 5. Cefepime/Zidebactam

Zidebactam is a new-generation diazobicyclooctane-derived inhibitor (DBO), non-β-lactam antibiotic, with a dual mode of action involving selective, high-affinity binding of the PBP2 of Gram-negative bacteria and inhibition of β-lactamases. Due to PBP2 binding, zidebactam alone demonstrates antibacterial activity against various isolates of *Enterobacterales* and *P. aeruginosa*. It has been shown that the combination of cefepime/zidebactam results in increased inhibitory activity and stability against the hydrolysis of a wide range of β-lactamases [62]. Zidebactam combined with cefepime in a 1:1 combination is in clinical development for the treatment of Gram-negative bacterial infections. Studies have evaluated the in vitro activity of cefepime combined with zidebactam against a large worldwide collection of contemporary clinical isolates of Gram-negative organisms [63], demonstrating potent in vitro activity against *Enterobacterales* and *P. aeruginosa*, including isolates producing all classes of clinically relevant β-lactamases (classes A, C, and D), except for MBLs. Zidebactam was shown to cause potentiation in vitro of cefepime against *S. maltophilia*, but modest potentiation occurred against *A. baumannii*, with elevated MIC values (≥16 mg/L) [64,65] (Table 1). For antimicrobial susceptibility testing purpose, zidebactam should be tested at a 1:1 concentration with cefepime [4]. No clinical breakpoint (CLSI, EUCAST, or FDA) has been approved for this combination. An EUCAST epidemiological cut-off (ECOFF) value has not been assigned.

Three phase 1 trials (NCT02707107, NCT02942810 and NCT03630094) [66,67,68] and one phase 3 trial (NCT04979806) [69] have defined the value of this molecule. The efficacy, safety, and tolerability of cefepime/zidebactam (2 g of cefepime plus 1 g of zidebactam, every 8 h) in comparison to meropenem (1 g every 8 h) in the treatment of hospitalized patients with cUTIs or acute pyelonephritis are being examined as part of a phase 3 randomized, double-blind, multicenter, non-inferiority study (NCT04979806) [69]. A total of 504 hospitalized adults (≥18 years of age) with cUTIs or acute pyelonephritis will participate in the research project. A combination of clinical symptoms and signs and the presence of pyuria will be used to diagnose cUTIs or acute pyelonephritis. The research drugs’ treatment period lasts between seven and ten days [69]. No study results are posted on ClinicalTrials.gov for this study at this time, but are expected after August 2022 (Table 2).

The PKs of the two compounds are similar: in healthy volunteers, cefepime has a Vd of 15.4 ± 2.9 L and a PB of 20% while zidebactam has a Vd of 17.4 ± 3.2 L and a PB of <15% [69]. Data regarding the tissue penetration are available only for the respiratory tract: the ELF to total plasma penetration ratio, after multiple doses of cefepime 2 g plus 1 g zidebactam every 8 h, is 39% and 38%, respectively, while alveolar macrophage to total plasma ratios are 27% and 10%. The penetration ratio is based on total plasma concentration, as both agents have low plasma PB [20]. In healthy volunteers, cefepime has a mean t½ of 2.0 (±0.2) h and a Cl of 6.36 (±1.35) L/h, whereas zidebactam has a t½ of 1.9 (±0.3) and a Cl of 7.44 (±1.54) [20]. Both compounds are renally eliminated and dosage adjustments are required in patients with renal failure [70] (Table 3 and Table 4).

## 6. Cefiderocol

Cefiderocol is a combination of a catechol-type siderophore and a cephalosporin core with side chains similar to cefepime and ceftazidime. A catechol moiety on the 3-position of the R2 side chain allows cefiderocol to function as a siderophore molecule, chelating extracellular iron. Following the chelation of iron, cefiderocol is transported to the periplasmic space through ferric iron transport systems located on the outer membrane of Gram-negatives. Once within the periplasmic space, cefiderocol dissociates from the iron and binds to PBPs, inhibiting peptidoglycan cell wall synthesis [71]. Its unique structure and mechanism of action confer enhanced stability against hydrolysis by many β-lactamases, such as CTX-M, and carbapenemases KPC, NDM, VIM, IMP, OXA-23, OXA-48-like, OXA-51-like, and OXA-58 [72]. Cefiderocol has a broad antibacterial spectrum against a variety of aerobic bacteria, including *Enterobacterales*, *Acinetobacter* spp., *Pseudomonas* spp., *Burkholderia* spp., and *S. maltophilia*. Isolates of *A. baumannii* producing PER-like β-lactamases and NDM-like β-lactamases showed reduced susceptibility to cefiderocol. Malik et al. reported that reduced expression of the siderophore receptor gene *pir*A is correlated with resistance to cefiderocol in *A. baumannii*. Moreover, mutations involving the PBP3 may also contribute to cefiderocol resistance [73]. Interestingly, cefiderocol in combination with avibactam exhibited excellent activity against all OXA-23 and PER-like β-lactamase coproducing isolates [74] (Table 1). Cefiderocol formulation is commercially available (1 g vials). EUCAST provided a clinical breakpoint of ≤2 mg/L for *Enterobacterales* and *Pseudomonas* spp., while CLSI provided a clinical breakpoint of ≤4 mg/L for *Enterobacterales*, *P. aeruginosa*, *Acinetobacter* spp., and *S. maltophilia* [4].

A randomized, open-label, prospective, phase 3 clinical trial for cefiderocol was conducted in patients with carbapenem-resistant Gram-negative bacterial infections, regardless of species or source of infection, including sepsis and BSIs (ClinicalTrials.gov registration: NCT02714595) [38]. As defined by the best available therapy, clinical cure rates in nosocomial pneumonia (NP; 50%) and BSI (53%) were comparable between cefiderocol and the comparator (43 vs. 43%) [38]. In CREDIBLE-CR, cefiderocol was associated with favorable microbiological outcomes vs. the best available therapy when it came to cUTIs (53 vs. 20%) [38]. Moreover, cefiderocol caused a higher number of deaths, particularly in the *Acinetobacter* spp. subgroup, a finding for which no clear explanation was offered [38]. APEKS-NP is a randomized, double-blind, phase 3, non-inferiority investigation published recently by Wunderink et al. [75]. This study included 148 participants who were given cefiderocol and 152 subjects who were given meropenem. Cefiderocol was found to be non-inferior to high-dose extended-infusion meropenem in patients with Gram-negative NP, and mortality on day 14 was similar in all groups (12.4 vs. 11.6%) [75]. Moreover, Hsueh et al. examined the cefiderocol, ceftolozane/tazobactam, and ceftazidime/avibactam microbiological profiles in vitro for *P. aeruginosa, S. maltophilia* and *A. baumannii* bloodstream isolates [76] (Table 2). *P. aeruginosa* isolates resistant to colistin and imipenem were more susceptible to cefiderocol in vitro than those resistant to ceftolozane/tazobactam and ceftazidime/avibactam [76]. 

Specific tissue penetration data are available for the respiratory system: in healthy volunteers after a single 2 g i.v. dose, the drug penetrates into ELF with geometric mean concentration ratios, over 6 h, ranging from 0.0927 to 0.116 mg/L for ELF and total plasma [77]. In patients with VAP, the geometric mean ELF concentration of cefiderocol was 7.63 mg/L at the end of infusion, and 10.40 mg/L 2 h later. The ELF/unbound plasma concentration ratio was 0.212 (21.2%) at the end of infusion and 0.547 after 2 h, suggesting delayed lung distribution, with concentrations sufficient to treat Gram-negative bacteria [77]. 

Due to its hydrophilicity, cefiderocol shows urinary excretion with a negligible hepatic metabolism. After administration of multiple doses in healthy volunteers, the total drug Cl is 5.4 (±14.0) L/h, the t½ is 2.7 (±21.6) h [21] and the Vd 18 L (±3.36) [78]. Changes in renal function are the first cause of dose adjustment [79]; interestingly, this is true also for patients with CrCl >120 mL/min for which the administration interval should be reduced [80].

A population PK analysis, conducted on healthy volunteers and patients with cUTIs and uncomplicated pyelonephritis, demonstrated that the presence of infection is a significant covariate, which increases the Vd to 36% and Cl to 26% [81] (Table 3 and Table 4).

## 7. Ceftaroline/Avibactam

Ceftaroline-avibactam combines a fifth-generation broad-spectrum cephalosporin, with bactericidal activity against Gram-positive (including methicillin-resistant *S. aureus*) and Gram-negative pathogens, with avibactam, a diazabicyclooctane-derived molecule that can reversibly inhibit several β-lactamases, including Ambler class A, class C, and certain class D enzymes [82]. This association significantly extended the spectrum of action to ESBL-, AmpC-, KPC-, and OXA-48-producing *Enterobacterales*. More recently, it was demonstrated that ceftaroline-avibactam might increase the activity spectrum on *K. pneumoniae*-producing carbapenemases and multiple β-lactamases and modifications in OmpK35 and OmpK36 porins [83]. No activity was reported against MBL producers. Ceftaroline-avibactam demonstrated limited activity against *A. baumannii* and *P. aeruginosa* [84] (Table 1). For antimicrobial susceptibility testing purposes, the concentration of avibactam is fixed at 4 mg/L [4]. No clinical breakpoint (CLSI, EUCAST, or FDA) has been approved for this combination. An EUCAST epidemiological cut-off (ECOFF) value has not been assigned.

From a clinical point of view, ceftaroline-avibactam has been studied in a phase 2 trial (NCT01281462) in adults with cUTIs [85]. This study compared treatment with i.v.-co-administered ceftaroline fosamil, the prodrug of ceftaroline, and avibactam with different schedules, every 8 or 12 h; these arms were compared with doripenem or placebo. The study involved 217 patients and ended in 2012, and no results were posted or published elsewhere up to date. The chosen outcome measures were microbiological response to a test-of-cure and the safety profiles of the combined molecules. Secondary outcomes were the clinical response to the test of cure. Other completed works on that compound are mostly preclinical [NCT01624246, NCT01789528, NCT01290900] and results are not available [86,87,88]. Due to the interesting compound association, it would be of great interest not only to receive updates on this study, but also to observe from such a combination a potential development in the setting of HAP, VAP, ventilated hospital-acquired pneumonia (vHAP), and other complicated infections. No further conclusions can be extrapolated from the current available data on clinical use (Table 2).

In healthy volunteers, the PK parameters relative to the distribution of ceftaroline-fosamil/avibactam in the fixed-dose combination (600 mg/600 mg) are a Vd of 19.8 (±2.9) L when administered in a single dose and of 16.9 (±2.4) L when administered in multiple doses [23]. Ceftaroline-fosamil was approved in 2010 for the treatment of adults and children with community-acquired bacterial pneumonia and acute bacterial skin and skin structure infections [89]. In addition, the combination has been used off-label in serious infections such as nosocomial pneumonia, osteoarticular infections, meningitis, and endocarditis. In these tissues, ceftaroline-fosamil diffuses when administered at high doses (from 200 mg every 12 h to 800 mg every 8 h) [90] and has a good antibacterial effect. More studies are however needed to study the PK of the combination in more detail. Ceftaroline-fosamil/avibactam is primarily renally excreted; adjustment is therefore necessary for CrCL ≤ 50 mL/min [91] (Table 3 and Table 4). 

## 8. Ceftolozane/Tazobactam

Ceftolozane/tazobactam is a new β-lactam/β-lactamase inhibitor combination consisting of a fixed (2:1) combination of an antipseudomonal cephalosporin, ceftolozane, and a well-established β-lactamase inhibitor, tazobactam, approved by the FDA in 2014. The chemical structure of ceftolozane is based on oxyimino-aminothiazolyl cephalosporin with a pyrazole substituent at the 3-position side chain, instead of the lighter pyridium, typical of ceftazidime. This heavier side chain has the ability to penetrate through porin channels and provides a steric obstacle to hydrolysis mediated by ESBL and AmpC determinants [92]. Ceftolozane/tazobactam presents antimicrobial activity against *Enterobacterales* and *P. aeruginosa* by the inhibition of common class A β-lactamases (TEM, SHV, CTX-M) or of class C enzymes [93]. Ceftolozane/tazobactam antimicrobial activity is less affected by *P. aeruginosa* AmpC enzymes than ceftazidime/avibactam, and, for this reason, ceftolozane/tazobactam is commonly considered an antipseudomonal drug. According to the Italian survey on *P. aeruginosa*, ceftolozane/tazobactam was the most active anti-*Pseudomonas* agent; moreover, it was active against approximately half the isolates that are resistant to all other β-lactams or resistant to all other agents except colistin [94]. High levels of resistance to ceftolozane/tazobactam in *P. aeruginosa* were associated with the overexpression and structural modification of AmpC [95]. No antimicrobial activity has been reported against *A. baumannii* and *S. maltophilia*, as well as against carbapenemase-producing microorganisms [96]. Ceftolozane/tazobactam is commercially available in a 2:1 formulation (Table 1). For antimicrobial susceptibility testing purposes, the concentration of tazobactam is fixed at 4 mg/L [4]. EUCAST and CLSI provided a susceptibility clinical breakpoint of ≤2 mg/L for *Enterobacterales* and of ≤4 mg/L for *P. aeruginosa* [4].

For cUTIs and cIAIs, the FDA and EMA approved 1.5 g (ratio of 1.0 ceftolozane to 0.5 tazobactam) every 8 h, with a double dosage (3 g; 2:1 ratio of ceftolozane to tazobactam) approved for the phase 3 trial ASPECT-NP for nosocomial pneumonia [40,41,97]. Bassetti et al. described one of the largest clinical trials using ceftolozane/tazobactam in a multicenter cohort of 101 patients with documented *P. aeruginosa* infection [98]. The only independent predictor of clinical failure was sepsis for patients with clinical success in comparison to those who suffered clinical failure, according to multivariate analyses (OR = 3.02, 95% CI: 1.01–9.2; *p* = 0.05) [98]. CEFTABUSE-registered results showed a non-significant trend towards more favorable 14-day clinical cure rates in ceftolozane/tazobactam-treated patients than aminoglycoside or colistin-treated patients (81.3% vs. 56.3%; *p* = 0.11%) [99]. In addition, a systematic study concluded that ceftolozane/tazobactam therapy could be useful, even outside of an accepted setting of indication, for difficult-to-treat *P. aeruginosa* infections [100] (Table 2).

Lung penetration in healthy volunteers was 61% for ceftolozane and 63% for taniborbactam, assuming, respectively, a PB of 21% and 30% [101]. Similarly, in critically ill, mechanically ventilated patients, lung penetration was 50% and 62%, respectively [102]. Ceftolozane Vd is about 13.5 L, while tazobactam is 18.2 L; these parameters are increased in patients with pneumonia compared with healthy subjects [103] and in neonates, consistent with age-related physiologic changes [104,105]. The combination has been employed off-label in skin and soft-tissue, bone and joint, bloodstream and multiple infections, suggesting good tissue penetration also in these tissues [100]. However, the combination does not provide adequate exposure in cerebral spinal fluid [106]. Both compounds are renally eliminated and dosage adjustment is required in renal impairment (Table 3 and Table 4).

## 9. Ceftazidime/Avibactam

Ceftazidime/avibactam is a β-lactam/β-lactamase inhibitor combination available since 2015. Avibactam is structurally different from the other clinically used β-lactamase inhibitors since it does not contain a β-lactam core. The inhibitory mechanism proceeds by opening of the avibactam ring, but the reaction is reversible, because the deacylation leads to the regeneration of the compound and not to hydrolysis and turnover [107]. Compounding avibactam with ceftazidime resulted in overcoming resistance due to Ambler class A, class C, and some class D β-lactamases [11]. For this reason, ceftazidime/avibactam has become a first-line option against KPC- or OXA-48-producing *Enterobacterales*, and represents an alternative option against ESBL- or AmpC-producing *Enterobacterales* and against *P. aeruginosa*. Episodes of colonization or infection due to ceftazidime/avibactam resistant strains have rapidly been reported in the literature [108]. Resistance to ceftazidime/avibactam is commonly due to the presence of MBLs as their activity is not restored by avibactam. Other mechanisms include increased expression of the *bla*_KPC_ gene, specific mutations of genes coding for carbapenemases, changes in cell permeability (i.e., loss of porins), expression of efflux pumps, and, in the case of *P. aeruginosa*, by hyperexpression of PDC enzymes variants [109,110,111]. Ceftazidime/avibactam has no antimicrobial activity against *Acinetobacter baumannii* (no inhibition of *A. baumannii* OXA-type enzymes). Ceftazidime/avibactam is commercially available in a 4:1 formulation. For susceptibility testing purposes, the concentration of avibactam is fixed at 4 mg/L [112] (Table 1). EUCAST and CLSI provided a susceptibility clinical breakpoint of ≤8 mg/L for *Enterobacterales* and *P. aeruginosa* [4].

Ceftazidime/avibactam is approved for use in cIAIs, cUTIs, and HAP/VAP therapy, as well as in infections with microorganisms resistant to ceftazidime (RECLAIM, RECAPTURE 1&2, REPROVE, and REPRISE studies, respectively) [113,114,115,116]. Real-world data from patients with carbapenemases KPC and OXA-48 confirmed the clinical effectiveness of ceftazidime/avibactam [117,118,119,120]. Moreover, Fiore et al. found no differences in mortality rates between ceftazidime/avibactam monotherapy and combination therapy (N = 503 patients; direct evidence OR: 0.96; 95% CI: 0.65–1.41), and Onorato et al. found similar results in an unpublished systematic review [121,122]. A retrospective longitudinal investigation of 138 patients with KPC-producing *K. pneumoniae* bacteremia, whose mortality was considerably lower than that of a matched cohort of patients treated with medications other than ceftazidime/avibactam (36.5% against 55.8%, *p* = 0.005), was addressed by Tumbarello et al. [118]. Moreover, Shields et al. found that treatment with ceftazidime/avibactam had a significantly greater rate of clinical success (85 vs. 48/40/37%, *p* = 0.02) and survival at 90 days (92 vs. 69/55%, *p* =0.01) than other regimens, as well as higher renal safety compared to aminoglycoside- and colistin-based regimens [120] (Table 2).

PK studies have shown that the approved dosage (2/0.5 g every 8 h) provides adequate plasma levels [8] and sufficient drugs’ distribution in all approved indications [123]. In particular, the ELF:plasma penetration ratios are 52% for ceftazidime and 42% for avibactam. Cefepime-avibactam has been successfully used in the treatment of serious infections with limited treatment options and in tissues with difficult drug penetration [124], such in bone and joint infections [125,126], endocarditis [127], mediastinitis [128], abscesses [28], and post-transplant renal necrosis [129]. Ceftazidime PB is approximately 10% and the Vd is 14.3 L [130]. Avibactam PB is also low (5.7–8.2%) and the Vd is approximately 15–25 L [29]. Both drugs have a t½ of approximately 2 h [11] and dose adjustment is required in patients with moderate and severe renal impairment [124]. In patients on hemodialysis, the dose has to be administered after hemodialysis (Table 3 and Table 4). 

## 10. Imipenem/Relebactam

Relebactam is a non-β-lactam, bicyclic diazabicyclooctane, β-lactamase inhibitor, structurally related to avibactam, but differing by the addition of a piperidine ring to the 2-position carbonyl group. Both inhibitors display activity against Ambler class A and class C β-lactamases [108,131,132]. Imipenem/relebactam has improved activity against *P. aeruginosa* with decreased expression of OprD and overproduction of AmpC β-lactamases, thanks to relebactam AmpC inhibition. Imipenem/relebactam maintains a limited activity against *bla*_OXA-48_-expressing carbapenem-resistant *Enterobacterales*, and has no activity against MBLs (including IMP, VIM, and NDM)-producing isolates. Relebactam has no activity against OXA class D β-lactamases of *A. baumannii*. Based on the data available, the addition of relebactam does not improve the activity of imipenem against *A. baumannii* and *S. maltophilia* [133,134]. Imipenem/relebactam is now commercially available in a 1:1 formulation (plus cilastatin). For susceptibility testing purposes, the concentration of relebactam is fixed at 4 mg/L [112] (Table 1). EUCAST provided a susceptibility clinical breakpoint of ≤2 mg/L for *Enterobacterales*, *P. aeruginosa*, and *Acinetobacter* spp., while CLSI provided a susceptibility clinical breakpoint of ≤1 mg/L for *Enterobacterales* and ≤2 mg/L for *P. aeruginosa* [4,112].

Two randomized, controlled, comparative, phase 3 clinical trials on imipenem/relebactam, RESTORE-IMI 1 and RESTORE-IMI 2, were conducted. In the first, the efficacy and safety of imipenem/relebactam was comparable to colistin plus imipenem for the treatment of imipenem-non-susceptible bacterial infections (including cIAIs, cUTIs, HAP and VAP in 47 patients with 16% *K. pneumoniae* and 16% KPC), with a 70% favorable overall response. A significantly lower incidence of nephrotoxicity was reported for imipenem/relebactam (10% vs. 56%, *p* = 0.002) [135]. In the second study, imipenem/relebactam was found to be non-inferior to piperacillin/tazobactam for the treatment of HAP/VAP. A sample of 537 patients was enrolled, and empiric linezolid administered in both arms. Overall, the data showed a favorable profile for imipenem/relebactam for critically ill and high-risk patients. Reported adverse events in both studies warned of the potential gastrointestinal disturbances, electrolyte imbalances, phlebitis and/or infusion-site reactions, fever, headache, and hypertension [136]. A phase 3 non-randomized, not controlled, open-label clinical study investigated the safety and efficacy of imipenem/relebactam in 81 Japanese subjects with cIAIs or cUTIs (14 bacteremic, 7 septic). Microorganisms were mostly non-MDR, and the results were in line with registration studies, showing comparable favorable efficacy and safety [137]. The microbiological features of the study may not add informative data for MDR pathogens. Finally, a phase 4 investigator-initiated, open-label, randomized, single-center trial is recruiting participants to study the clinical response of imipenem/relebactam in febrile neutropenia (NCT04983901) [45] (Table 2).

Because of their hydrophilic structures, the distribution of imipenem/relebactam is prevalent in the interstitial spaces; PB is about 20% for imipenem, 20% for cilastatin and 22% for relebactam; Vd is 24.3 L for imipenem and cilastatin and 19 L for relebactam [138]. The two drugs achieve relatively high concentrations in the respiratory system: the exposure in ELF, relative to that of unbound concentrations in plasma, is 55% for imipenem and 54% for relebactam [31]. As expected, imipenem was not detected in alveolar cells, providing further confirmation that its concentrations in the extracellular compartment are relevant for treating pneumonia [31]. Both imipenem and relebactam have renal Cl and a t½ of approximately 1 h [[138]; dose adjustment should be performed in renal impairment [139]. In hemodialyzed patients, the dose has to be administered after hemodialysis (Table 3 and Table 4). 

## 11. Meropenem/Nacubactam

Similarly to zidebactam, nacubactam belongs to a new generation of DBO inhibitor. The meropenem/nacubactam combination exerts a potentiated spectrum of activity against class A, C, and some class D β-lactamases (a weak interaction with subclass 2 d enzymes), and promotes a further affinity for PBP2. This combination may potentially overcome ceftazidime/avibactam-resistant isolates among KPC-producing *Enterobacterales* due to mutation in the Ω-loop, with MIC values ≤ 8 mg/L [132,140]. The enhanced activity of the meropenem/nacubactam combination was demonstrated against class A serine carbapenemase-producing *Enterobacterales* [141], and against meropenem-resistant *P. aeruginosa* clinical isolates [142]. No data are available for activity against OXA-48-producing *Enterobacterales*. This combination showed no enhanced activity in comparison to meropenem alone, against *A. baumannii* [143]. Meropenem/nacubactam is currently in phase 2 trial (Table 1). For antimicrobial susceptibility testing purpose, nacubactam should be tested at a 1:1 concentration with meropenem [4]. No clinical breakpoint (CLSI, EUCAST, or FDA) has been approved for this combination. An EUCAST epidemiological cut-off (ECOFF) value has not been assigned.

To date, no clinical studies have explored the real-life use of the combination of meropenem/nacubactam in either phase 2 or 3 trials. The only study that explored the topic deeply is a non-randomized, open-label, one-treatment, one-group study in participants with cUTIs, including pyelonephritis, to characterize the PK of nacubactam co-administered with meropenem (NCT03174795) [144]. The study involved 20 patients and ended in 2017. The results were not displayed, and the primary outcomes were mainly pharmacokinetically directed. In preclinical studies, the chosen dosage for meropenem was similar to that for meropenem/vaborbactam, for which superposable considerations could be extrapolated. The addition of nacubactam may lead to wider microorganism coverage, as discussed in the previous section. We shall await clinical studies to broaden these considerations before making more inferences into the clinical use of this promising molecule (Table 2).

Two phase 1 studies show that the coadministration of 2000 mg of meropenem and 2000 mg of nacubactam does not significantly alter the PKs of either drugs [33]; the two compounds show similar PK after a single i.v. administration: meropenem shows a Vd of 15–20 L, a t½ of 1 h and a low PB of 2%, while for nacubactam, a Vd of 21.9 L, a t½ of 2.66 h, and a PB of 2% have been demonstrated [33]. 

A clinical study to investigate the intrapulmonary lung penetration of the combination in healthy volunteers has been completed, but results have not been posted yet [145]. Pre-clinical studies have been conducted in neutropenic murine models: after administering a dose mimicking the combination dosage of 2000/2000 mg every 8 h in humans, the %t > ELF at different drug concentrations and AUC_0–24_ were comparable in humans and mice, validating the animal model to assess the efficacy of the combination [142] (Table 3 and Table 4). 

## 12. Meropenem/Vaborbactam

Meropenem/vaborbactam is a novel carbapenem-boronic acid β-lactamase inhibitor formulation approved by the FDA in 2017 [146,147]. Vaborbactam was designed to improve the performance of meropenem against carbapenemase-producing organisms. The boronic structure of vaborbactam forms a reversible covalent bond with the catalytic serine site of the β-lactamases [148]. Meropenem/vaborbactam presents antimicrobial activity against class A and class C β-lactamase-producing *Enterobacterales,* especially those producing ESBL, KPC, and AmpC determinants; hence, it represents a first-line choice for the treatment of KPC-producing *Enterobacterales*. Meropenem/vaborbactam was also shown to be active against strains of *Enterobacterales* producing other types of class A serine carbapenemases, such as SME and NMC-A enzymes [147]. Resistance to meropenem/vaborbactam in KPC-producing *Enterobacterales* is currently very rare and mostly due to porin inactivation (OmpK35/36) [149,150,151,152]. Interestingly, meropenem/vaborbactam retains activity also against strains producing KPC mutants that confer resistance to ceftazidime/avibactam (e.g., KPC-8, KPC-31) [153]. Accordingly, meropenem/vaborbactam is more specific than ceftazidime/avibactam against KPC-producing *Enterobacterales*. The activity of meropenem/vaborbactam against *P. aeruginosa* and *A. baumannii* was found to be similar to that of meropenem alone. In fact, in these species, meropenem resistance is largely mediated by mechanisms that are not antagonized by vaborbactam (e.g., outer-membrane impermeability, upregulation of efflux systems, and production of class B or class D β-lactamases) [154]. No antimicrobial activity has been reported for MBL-producing Gram-negatives and OXA-48-producing *Enterobacterales*. Meropenem/vaborbactam is commercially available in a 1:1 formulation. For susceptibility testing purposes, the concentration of vaborbactam is fixed at 8 mg/L [112] (Table 1). EUCAST provided a susceptibility clinical breakpoint of ≤8 mg/L for *Enterobacterales* and *P. aeruginosa*, while CLSI provided a susceptibility clinical breakpoint of ≤4 mg/L only for *Enterobacterales* [112].

The efficacy, tolerability, and safety of meropenem/vaborbactam for the treatment of cUTIs and acute pyelonephritis have been investigated in a phase 3 non-inferiority trial (TANGO I) [47]. In this study, 59.1% of patients were diagnosed with acute pyelonephritis and 40.9% with cUTIs. The most common pathogens were *Enterobacteriaceae* (29% ESBL) and *P. aeruginosa*. Per-pathogen clinical outcomes and microbiological eradication rates were similar among treatment groups. Meropenem/vaborbactam was found to be non-inferior to piperacillin/tazobactam for the primary outcome. Thereafter, a randomized, open-label trial investigated patients with cUTIs, HAP/VAP, bacteremia or cIAIs due to known or suspected carbapenem-resistant *Enterobacteriaceae*, of whom 63% were KPC-producing (TANGO II) [155]. The comparator was the “best available therapy” (including a variety of molecules, 67% in combination), and the results showed the superiority of meropenem/vaborbactam, especially in immunocompromised patients. The trial was terminated prematurely after an interim analysis demonstrating higher cure rates and lower mortality and nephrotoxicity rates with meropenem/vaborbactam. Lastly, NCT03006679, a phase 3b, double-blind, multicenter study, was launched to compare meropenem/vaborbactam with piperacillin/tazobactam for the treatment of HAP/VAP. The study was withdrawn due to the sponsor’s decision [156]. Preliminary real-world experiences have been published and have shown good results and confirmed data from trials, specifically with regard to carbapenem-resistant *Enterobacteriaceae* and KPC-producing bacteria and in isolates resistant to ceftazidime/avibactam [157,158,159,160]. Caution should be used in patients treated with valproic acid for potential interactions (Table 2).

The association is approved only for the treatment of cUTIs [34,161], at the combination ratio 2000/2000 mg q 8 h [[162]; however, Wenzler et al. found that its intrapulmonary penetration, based on the AUC (0–8) ratio of ELF and unbound plasma concentration, was 65% for meropenem and 79% for vaborbactam [35], suggesting a potential role on HAP and VAP [163]. Adequate penetration into cerebrospinal fluid, interstitial space, and tissue compartments has been demonstrated for meropenem; however, no data are currently available on vaborbactam or the association of the two drugs in these districts. Meropenem PB is approximately 2% and the Vd is 20.2 L, vaborbactam PB is instead 33% and the Vd is 18.6 L. The plasma Cl of meropenem and vaborbactam are similar. The t½ is 1.3 and 1.9 h, respectively; both drugs are renally eliminated and dosage adjustments are needed in patients with renal impairment [35,162]. In hemodialyzed patients, the dose has to be administered after hemodialysis [34] (Table 3 and Table 4).

## 13. Conclusions

Infections sustained by MDR or extensively drug-resistant (XDR) Gram-negatives represent a serious cause of concern. A “standard of care” for these infections is lacking, therefore familiarity with clinical, microbiological, and PK/PD data of new molecules/compounds is fundamental to achieve better clinical outcomes. In this review, we provided microbiological, clinical and pharmacological data for the new BL/BLIs and cefiderocol, to be used as an “all-inclusive” guide for clinicians to counsel the proper antibiotic therapy against infections sustained by MDR or XDR pathogens. It is of note that some less well-represented pathogens, such as *Burkholderia* spp., *Pandorea* spp., *Elizabethkingia* spp., *Chryseobacterium* spp. and *Myroides* spp. that usually have MDR or XDR phenotypes, may not be susceptible to most antibiotics here presented. This aspect did not reflect the aims of this review and was not included, but does warrant more specific studies.

Considering microbiological targets, most new antibiotics are active against ESBL, AmpC, and OXA-48-like determinants, while only a few showed antimicrobial activity against MBL producers (including *S. maltophilia*) and *A. baumannii*, thereby still representing an important challenge for the treatment of infectious diseases. Notably, for *P. aeruginosa*, good therapeutic options are currently or potentially available. Cefiderocol presents a broader spectrum of antimicrobial activity, being active against all investigated targets. Conversely, ceftolozane/tazobactam presents the most restricted antimicrobial activity, being active only against ESBL, AmpC, and *P. aeruginosa*. In addition, aztreonam/avibactam, cefepime/taniborbactam, and cefepime/zidebactam present wider antimicrobial activity, being active against six or seven investigated targets; hence, they are deserving of high expectations for their future introduction in clinical therapy. 

In-depth knowledge of PK and PD properties of these antibiotics or antibiotic combinations is warranted to optimize prescribing and to preserve their antibacterial activity. From a PK point of view, all the mentioned agents are hydrophilic drugs and their relative solubility impacts on their volume of distribution, which mostly corresponds to extracellular fluids. The degree of plasma PB is also important, since only unbound drugs are able to exert antimicrobial activity; with the exception of cefiderocol with a PB of 40–60%, all other compounds are usually characterized by a low-to-moderate plasma PB. On the basis of these characteristics, these antibiotics diffuse easily in tissues and reach antibacterial levels in the ELF, therefore being useful in pulmonary infections. Finally, the hydrophilic nature of β-lactams is responsible for the route of elimination; for these agents, it is almost always renal. Efficacy in cICUs has been demonstrated; however, dose adjustment in patients with renal insufficiency is required.

β-Lactams exhibit time-dependent antibacterial effects, and maintaining the unbound drug concentration above the MIC (ft > MIC) for a significant part of the dosing interval predicts microbiological efficacy. This is particularly important in critically ill subjects [164,165] and in infections caused by multidrug-resistant bacteria [166]. Indeed, the inability to attain fT > MIC > 50% has been associated, in a large study in critically ill patients, with a 32% decreased likelihood of a positive clinical outcome [167] and, to attain this target, continuous infusion regimens have been proposed [168,169,170]. For most compounds described in the present review, maintenance of fT >MIC of 40–50% or more has been suggested, with longer values described only for cefiderocol (Table 3) and continuous infusion is therefore employed; however, for certain drugs, including the carbapenems and some of the newest cephalosporins such as ceftaroline, stability at room temperature is limited to 3–4 h and therefore these agents are better administered as a prolonged infusion to enhance pharmacodynamic exposure while retaining stability requirements.

From a clinical point of view, most studies were performed in patients with cICUs and fewer in patients with BSIs. Cefiderocol, ceftazidime/avibactam, and meropenem/vaborbactam are the compounds with broader clinical indications, although the microbiology spectrum is different with cefiderocol having the broadest one, followed by ceftazidime/avibactam and meropenem/vaborbactam. The “pro and cons” of the different new antibiotic compounds are shown in Table 5.

Belonging to the class of β-lactams, which inhibit the synthesis of the bacterial peptidoglycan cell wall, a bacterial target absent in eukaryotic cells, these agents have low direct toxicity. Hypersensitivity reactions are the most common adverse effect and, because of the common β-lactam ring, cross reactivity can occur. In addition, allergic reactions can be directed against the side chain; for instance, the R1 side chain is identical in cefotaxime, cefiderocol, and aztreonam, and cross reactivity between these agents has been described [171]. Central nervous system dysfunctions with headache, confusion, and seizure risk have been also described for all the compounds and associations, particularly in patients treated with high doses or with renal dysfunction. Finally, all these agents can change the composition of the microflora of the gastrointestinal tract, and *Clostridioides difficile* infections are a possible risk.

In conclusion, in consideration of their spectrum of activity, PK/PD characteristics, and relative low toxicity, these compounds represent an interesting possibility in the treatment of MDR or XDR Gram-negative bacteria. However, despite the recent introduction of these antibiotics, resistance has already been reported (especially for ceftazidime/avibactam). Known potential resistance mechanisms to the described antibiotic compounds are summarized in Table 6.

This review suggests that a cautious and optimal antimicrobial stewardship, also considering combination therapy including old and new molecules, is strongly advisable, in order to preserve last-resort antibiotics.

## Figures and Tables

**Table 1 pharmaceuticals-15-00463-t001:** Microbiological targets.

	ESBL	KPC	MBL	AmpC	OXA-48	*P. aeruginosa* (MDR/XDR)	*Acinetobacter* (MDR/XDR)	*S. maltophilia*
Aztreonam/avibactam								
Cefepime/enmetazobactam								
Cefepime/taniborbactam								
Cefepime/zidebactam								
Cefiderocol								
Ceftaroline/avibactam								
Ceftolozane/tazobactam								
Ceftazidime/avibactam								
Imipenem/relebactam								
Meropenem/nacubactam								
Meropenem/vaborbactam								

Green = antimicrobial activity, red = no antimicrobial activity, yellow = partial antimicrobial activity, grey = not available. ESBL = extended-spectrum β-lactamase, Ambler Class A β-lactamases; KPC = *Klebsiella pneumoniae* carbapenemase, Ambler Class A β-lactamases; MBL = metallo-β-lactamases, Ambler Class B β-lactamases; AmpC = cephalosporinase, Ambler Class C β-lactamases; OXA-48 = oxicillinase-48, Ambler Class D β-lactamases; MDR = multidrug resistant; XDR = extended drug resistant.

**Table 2 pharmaceuticals-15-00463-t002:** Clinical settings investigated or under investigation for each compound.

	BSI	cIAI	cUTI/AP	HAP	VAP	Other (Limited Options)
aztreonam/avibactam						
cefepime/enmetazobactam						
cefepime/taniborbactam						
cefepime/zidebactam						
cefiderocol						
ceftaroline/avibactam						
ceftolozane/tazobactam						
ceftazidime/avibactam						
imipenem/relebactam						
meropenem/nacubactam						
meropenem/vaborbactam						

Green = existing data from clinical trials, red = clinical trials not performed or unavailable data. AP = acute pyelonephritis; BSI = β-lactamase inhibitors; cIAI = complicated intra-abdominal tract infection; cUTI = complicated urinary tract infection; HAP = hospital-acquired pneumonia; VAP = ventilator-associated pneumonia.

**Table 3 pharmaceuticals-15-00463-t003:** Pharmacokinetic parameters of β-lactams/β-lactamase inhibitors and cefiderocol. The concentrations of β-lactams and β-lactamase inhibitors were determined using liquid chromatography–tandem mass spectrometry.

DRUGS	PK/PD Index	T ½ (h)	Vd (L)	PB (%)	ELF/ Plasma (%)	References
aztreonam/avibactam	60% fT > MIC/ 50% fT > C_T_	2.3–2.8/1.8–2.2	20/26	56/8 *	30/30	[6,8,11,12,13]
cefepime/ enmetazobactam	60% fT > MIC/ 20–45% fT > C_T_	2.1/**	18.2/**	16–19/**	61/53	[14,15,16]
cefepime/taniborbactam	50% fT > MIC/ fAUC_24_/MIC	2.1/4.7 *	18.2/37.4	16–19/**	na	[16,17,18]
cefepime/zidebactam	30% fT > MIC/ fAUC_24/_MIC	2.0/1.9	15.4/17.4	20/< 15	39/38	[19,20]
cefiderocol	ƒT/MIC ≥75%	2.7	18	40–60	10–23	[11,21,22]
ceftaroline-fosamil/ avibacatm	40–50% fT > MIC/ f T > C_T_; fAUC	2.4/2.0 *	19.8/18 *	20/8 *	23/30 *	[8,23,24]
ceftolozane/ tazobactam	35% fT > MIC/ % f T > C_T_	3.5/2.5	13.5/18.2	21/30	61/63	[25,26,27]
ceftazidime/avibactam	50 % fT > MIC/ 40 % fT > C_T_	2.0/2.0	14.3/15–25	<10/5.7–8.2	52/42	[8,11,28,29,30]
imipenem/relebactam	6.5% fT > MIC/ fAUC_24_/MIC	1/1.2	24.3/19	20/22	55/54	[29,31,32]
meropenem/nacubactam	40% fT > MIC/ fAUC_24_/MIC *	1/2.6 *	15–20/21.9 *	2/2 *	na	[33]
meropenem/vaborbactam	40% fT > MIC/ fAUC_24_/MIC *	1.3/1.9	20.2/18.6	2/33	65/79	[34,35,36,37]

Abbreviations: *% f*T > MIC = percentage of time of unbound drug concentrations above MIC; *f*AUC_24_ = unbound drug area under the concentration time curve; MIC = minimum inhibitory concentration; C_T_ = critical concentration threshold; Vd = volume of distribution; T ½ = half-life; PB = protein binding; ELF = epithelial lung fluid. * No data for the combination are available yet; ** Data available for the β-lactam only.

**Table 4 pharmaceuticals-15-00463-t004:** Recommended dosages and dose adjustment in renal insufficiency.

Drugs	Recommended Dosage	Adjustment in RI	Authorized for Use in the European Union and by FDA	References
aztreonam/ avibactam	Not available	Not available	no	
cefepime/ enmetazobactam	Not available	Not available	no	
cefepime/ taniborbactam	Not available	Not available	no	
cefepime/ zidebactam	Not available	Not available	no	
cefiderocol	Pneumonia: 2 g q 8 h (7 days) cUTI: 2 g q 8 h (7–14 days)	CrCl ≥120 mL/min: 2 g q 6 h CrCl 60–120 mL/min: 2 g q 8 h CrCl 30–60 mL/min: 1.5 g q 8 h CrCl 15–30 mL/min: 1 g q 8 h CrCl <15 mL/min: 750 mg q 12 h	yes	[24,38,39]
ceftaroline-fosamil/ avibactam		Not available	no	
ceftozolane/ tazobactam	cIAI: 1.5–3 g q 8 h (4–5 days) Pneumonia: 3 g q 8 h (7 days) Bloodstream infection, skin and soft tissues: 1.5–3 g q 8 h cUTI: 1.5 g q 8 h	CrCl >50 mL/min: 1.5 g q 8 h 3 g q 8 h CrCl 30–50 mL/min: 750 mg q 8 h 1.5 g q 8 h CrCl 15–29 mL/min: 375 mg q 8 h 750 mg q 8 h	yes	[40,41,42,43,44]
ceftazidime/ avibactam	cIAI: 2.5 g q 8 (4–5 days) Pneumonia: 2.5 g q h (7 days) cUTI: 2.5 g q 8 h (5–14 days)	CrCl >50 mL/min: 2.5 g q 8 h CrCl 31–50 mL/min: 1.25 g q 8 h CrCl 16–30 mL/min: 0.94 g q 12 h CrCl 6–15 mL/min: 0.94 g q 24 h CrCl <5 mL/min: 0.94 g q 48 h	yes	[30]
imipenem/ relebactam	cIAI: 1.25 g q 6 h (4–7 days) Pneumonia: 1.25 g q 6 h (7 days) cUTI: 1.25 g q 6 h (5–14 days)	CrCl ≥90 mL/min: 1.25 g q 6 h CrCl 60–89 mL/min: 1 g q 6 h CrCl 30–59 mL/min: 0.75 g q 6 h CrCl 15–29 mL/min: 0.5 g q 6 h CrCl <15 mL/min: 0.5 g q 6 h	yes	[32,45,46]
meropenem/ vaborbactam	cUTI: 4 g q 8 h (5–14 days)	CrCl ≥50 mL/min: 4 g q 8 h CrCl 30–49 mL/min: 2 g q 8 h CrCl 15–29 mL/min: 2 g q 12 h CrCl <15 mL/min: 1 g q 12 h	yes	[37,47,48,49,50]
meropenem/nacubactam	Not available	Not available	no	

Abbreviations: CrCl = creatinine clearance, cIAI = complicated intra-abdominal tract infection; cUTI = complicated urinary tract infection; RI = renal insufficiency; FDA = US Food and Drug administration.

**Table 5 pharmaceuticals-15-00463-t005:** Pro and cons of new antibiotic compounds.

Antibiotic Compound	Pro	Cons
Aztreonam/avibactam	Good option against MBL bacteria	Uncertain activity against MDR PA
Cefepime/enmetazobactam	Option as “carbapenem sparing”	Activity limited to ESBL and AmpC
Cefepime/taniborbatam	Wide spectrum (including MBL)	Clinical data limited to cUTIs/AP
Cefepime/zidebactam	Wide spectrum (not including MBL)	Clinical data limited to cUTIs/AP
Cefiderocol	Very wide spectrum	Caution on *Acinetobacter* infections
Ceftaroline/avibactam	Spectrum covering also MRSA	Clinical studies limited to cUTIs/AP
Ceftolozane/tazobactam	Good data vs. *P. aeruginosa* pneumonia	Hydrolyzed by carbapenemases
Ceftazidime/avibactam	Good amount of clinical studies	Resistance is increasingly reported
Imipenem/relebactam	Good antipseudomonal activity	No clinical data on BSIs
Meropenem/nacubactam	Active against ESBL, KPC, and AmpC	Clinical data limited to cUTIs/AP
Meropenem/vaborbactam	Solid clinical studies against KPC	Not active against MBL, OXA-48, and MDR PA

AP: acute pyelonephritis; BSIs: bloodstream infections; cUTI: complicated urinary tract infection; ESBL: extended spectrum β-lactamases; MBL: metallo β-lactamases; MRSA: methicillin-resistant *Staphylococcus aureus*; PA: *Pseudomonas aeruginosa.*

**Table 6 pharmaceuticals-15-00463-t006:** Resistance mechanisms in target organisms.

Antibiotic	Target Organism	Resistance Mechanism
Aztronam/avibactam	*Enterobacterales*	Multiple β-lactamase production; mutations in PBP3 gene
	*P. aeruginosa*	Efflux, impermeability, PDC variants; presence of OXA determinants (other than OXA-48)
	*S. maltophilia*	Efflux; β-lactamase overexpression
Cefepime/enmetazobactam	*Enterobactarales*	NA
Cefepime/taniborbactam	*Enterobacterales*	NDM, VIM or IMP variants; impermeability
	*P. aeruginosa*	VIM variants; impermeability; PDC variants
	*S. maltophilia*	NA
Cefepime/zidebactam	*Enterobacterales*	Multiple β-lactamase production
	*P. aeruginosa*	Efflux; mutations in PBP genes
	*S. maltophilia*	NA
Cefiderocol	*Enterobacterales*	Mutations in genes involved in iron metabolism
	*P. aeruginosa*	Mutations in genes involved in iron metabolism; PDC variants
	*Acinetobacter*	Mutations in genes involved in iron metabolism; mutations in PBP genes
	*S. maltophilia*	Mutation in genes involved in iron metabolism
Ceftaroline/avibactam	*Enterobacterales*	NA
Ceftolozane/tazobatcam	*Enterobacterales*	Mutations in β-lactamase genes
	*P. aeruginosa*	Presence of GES or PER determinants; efflux, impermeability, PDC variants; overexpression of PDC
Ceftazidime/avibactam	*Enterobacterales*	Mutations in β-lactamase genes; efflux; β-lactamase overexpression; impermeability; multiple copies of β-lactamase genes
	*P. aeruginosa*	Efflux, impermeability, PDC variants
Imipenem/relebactam	*Enterobacterales*	Impermeability
	*P. aeruginosa*	Efflux; impermeability
Meropenem/nacubactam	*Enterobacterales*	NA
	*P. aeruginosa*	NA
Meropenem/vaborbactam	*Enterobacterales*	Efflux; impermeability; multiple copies of β-lactamase genes

NA: not available; PDC: Pseudomonas-derived cephalosporinase.

## Data Availability

Data sharing not applicable.

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
