# Peer review of "Microbiological, Clinical, and PK/PD Features of the New Anti-Gram-Negative Antibiotics: β-Lactam/β-Lactamase Inhibitors in Combination and Cefiderocol—An All-Inclusive Guide for Clinicians"

_pharmaceuticals, 2022, doi:10.3390/ph15040463_

Round 1
Reviewer 1 Report
The manuscripts provide an overview of data on the following compounds: aztreonam/avibactam, cefepime/enmetazobactam, cefepime/taniborbactam, cefepime/zidebactam, cefiderocol, ceftaroline/avibactam, ceftolozane/tazobactam, ceftazidime/avibactam, imipenem/relebactam, meropenem/nacubactam and meropenem/vaborbactam. This paper provides a lot of experimental information and provides a basis for following research. Especially, figure 1, figure 2 and table 1 are very good and useful.
There are still some problems in the article:
- The structures of the compounds should be provided.
- For PK/PD review, the information of the analysis instruments, and the analytical method should be discussed in this manuscript.
- A section of the introduction should be added in the beginning of the article, providing the background information.
Author Response
The manuscripts provide an overview of data on the following compounds: aztreonam/avibactam, cefepime/enmetazobactam, cefepime/taniborbactam, cefepime/zidebactam, cefiderocol, ceftaroline/avibactam, ceftolozane/tazobactam, ceftazidime/avibactam, imipenem/relebactam, meropenem/nacubactam and meropenem/vaborbactam. This paper provides a lot of experimental information and provides a basis for following research. Especially, figure 1, figure 2 and table 1 are very good and useful.
There are still some problems in the article:
The structures of the compounds should be provided.
We thank the reviewer for the suggestion and added Figure 1 with the chemical structures of the compounds.
For PK/PD review, the information of the analysis instruments, and the analytical method should be discussed in this manuscript.
For pharmacokinetic studies the concentrations of ß-lactams and ß-lactamase inhibitors were determined using liquid chromatography-tandem mass spectrometry. This information has been added in the legend to Table 3.
A section of the introduction should be added in the beginning of the article, providing the background information.
Thank you, an introduction paragraph has been added.
Reviewer 2 Report
The review article “Microbiological, clinical and PK/PD features of the new anti 2 Gram-negative antibiotics: β-lactam/β-lactamase inhibitors and 3 cefiderocol” is nicely written. The author try to cover the recent details about new BL/BLI and cefiderocol compounds. I have some important suggestions which will improve the quality of the paper:
Authors need to structure the manuscript properly in a journal format. To do that, start with the introduction about new BL/BLI and cefiderocol, their mode of action, their limitations, and challenges before jumping on details of each compound. You can also add an overall discussion in the end, about all these compounds in one paragraph, following the conclusion.
Authors should also include minimum 2 more figures, either to show the action mechanisms of these compounds on bacteria cells or how bacteria develop resistance against new BL/BLI and cefiderocol, by following different pathways, etc. Without figures, the manuscript loses the interest of readers.
Figures 1 & 2 are not figures, and they are tables. Please change accordingly.
Authors can advise any possibility to overcome these challenges described in the paper.
Thanks
Author Response
The review article “Microbiological, clinical and PK/PD features of the new anti 2 Gram-negative antibiotics: β-lactam/β-lactamase inhibitors and 3 cefiderocol” is nicely written. The author try to cover the recent details about new BL/BLI and cefiderocol compounds. I have some important suggestions which will improve the quality of the paper:
Authors need to structure the manuscript properly in a journal format. To do that, start with the introduction about new BL/BLI and cefiderocol, their mode of action, their limitations, and challenges before jumping on details of each compound. You can also add an overall discussion in the end, about all these compounds in one paragraph, following the conclusion.
Thank you, an introduction paragraph has been added.
Authors should also include minimum 2 more figures, either to show the action mechanisms of these compounds on bacteria cells or how bacteria develop resistance against new BL/BLI and cefiderocol, by following different pathways, etc. Without figures, the manuscript loses the interest of readers.
Thank you. We added a table with “pro and cons'' of different antibiotic compounds. Since all new antibiotics (here described) have almost the same mechanism of action (inhibition of cell wall by acting on PBPs), except for cefiderocol that has a peculiar mechanism of penetration in bacterial cells, we prefer to present their mechanisms in detail along each paragraph and not by figure. Moreover, to meet the Reviewer’s request, we added a table with the main recognized resistance mechanisms for each compound.
Figures 1 & 2 are not figures, and they are tables. Please change accordingly.
Thank you. We modified it
Authors can advise any possibility to overcome these challenges described in the paper.
Thanks
Reviewer 3 Report
The review manuscript presented by Principe et al. summarizes available literature data regarding β-lactam/ β-lactamase inhibitors due to problem of emerging bacterial resistance. The provided overview of data seems interesting and may be very useful, especially to the clinicians and physicians in general. Please find attached the comments.

Author Response
The review manuscript presented by Principe et al. summarizes available literature data regarding β-lactam/ β-lactamase inhibitors due to problem of emerging bacterial resistance. The provided overview of data seems interesting and may be very useful, especially to the clinicians and physicians in general. However, the present form the review contains several shortfalls.
The manuscript title “Microbiological, clinical and PK/PD features of the new anti Gram-negative antibiotics: β-lactam/β-lactamase inhibitors and cefiderocol” should be reformulated in order to clearly represent its content. The manuscript mainly deals with β-lactam/β-lactamase inhibitors in combinations. Thus in the title that fact should be stated (e.g. Microbiological, clinical and PK/PD features of the new anti Gram-negative antibiotics: β-lactam/β-lactamase inhibitors in combination and cefiderocol).
The introduction chapter is missing. Please provide short description of current world wide situation regarding β-lactam/β-lactamase inhibitors and bacterial resistance to them.
Thank you, an introduction paragraph has been added.
Page 16, lines 571-574: Authors stated that review provide data as an “all-inclusive” guide to counsel the proper antibiotic therapy. If so, the minimal inhibitory concentrations (MICs) values and recommended dose of application for each antibiotic described in the manuscript should be given. As well as, MICs of the processed antibiotics in combinations, if available.
Thank you for your comments. Breakpoint MIC values (when existing) and dosages (when available) have been specified for each antibiotic.
When available, recommended doses for each antibiotic have been added in Table 4.
Page 18, lines 631-633: This sentence is not suitable for conclusion. I suggest including this valuable information in the table for all antibiotics mentioned in the manuscript (e.g. authorized for use in EU, and/or other countries). It would be good to include this information for whole world. As well as, information regarding its toxicity, spectrum of activity (general classification: narrow-spectrum antibiotics as more specific and only active against certain groups or strains of bacteria vs. broad-spectrum antibiotics instead inhibit a wider range of bacteria, or even better make some new, your own more precise classification). In general, a as far as I am concerned the table with pros and cons information about β-lactam/ β-lactamase inhibitors as anti Gram –negative antibiotics and some final recommendation is lacking. Such table will significantly improve the quality of the present form of the manuscript.
We thank the reviewer, and, as suggested, we deleted the sentence on EU authorization from Conclusion. The information, together with advice on FDA approval, has been added in Table 4.
A “pro and cons” table has been added (Table 5).
Known potential resistance mechanisms to the described antibiotic compounds have been summarized in Table 6.
Minor comments:
Figures 1 and 2 according to my opinion are tables.
Thank you. We modified it accordingly.
Page 4, Table 1: The PK/PD index is not transparently presented. It is hard to track which value goes for which row, please make it more uniform (e.g. values distributed max in two rows).
We corrected the table according to the Reviewer’s suggestions, now the PK/PD index values are distributed in two rows.
Page 8, line 231-231: Provide the explanation of the abbreviation when for the first time mentioned in the text. E.g. First is stated “PB is 20%”, and then “a protein binding <15 %”. This should be stated when “plasma protein binding” was mentioned for the first time in the manuscript. The same goes for others abbreviations.
Thank you, all abbreviations have been corrected.
Pages 4 and 5: Regardless the text of the manuscript, please provide a proper explanation for each abbreviation used in the tables, e.g. Table 1. Vd – volume of distribution, T1/2 – half life, PD – protein binding; Table 2. RI etc.
Thank you, abbreviations used in tables have been explained in the legend.
Page 4, Table 1: measurement unit for Vd should be in brackets.
Done.
Page 5, Table2: Avoid the abbreviations in titles of tables and/or manuscript section titles
Thank you. It has been corrected
Page 9, Line 278: “in vitro” Latin words should be written in italic font.
Thank you. It has been corrected
Page 17, lines 603 and 609: Please correct ft>MIC to fT>MIC
Thank you. It has been corrected
Round 2
Reviewer 2 Report
Dear Editor,
I am convinced with the changes made.
Thank you
Author Response
We sincerely thank the reviewer.
Reviewer 3 Report
Dear authors, congratulations on the comprehensive review. You did a great job. Hope that other researchers will find it very interesting and useful, as it was to me.
I have few minor comments.
The title of the table should be written before (above) the table.
Table 5. P. aeruginosa and Acinteobacter should be written in italic, as well as on page 4, line 99 K. pneumoniae.
Author Response
The title of the tables have been moved above the table.
Table 5. P. aeruginosa and Acinteobacter have been written in italic, as well as on page 4, K. pneumoniae.
We tank again the reviewer for the comments.